# Ionizing Radiation Selectively Increases CXC Ligand 10 Level via the DNA-Damage-Induced p38 MAPK-STAT1 Pathway in Murine J774A.1 Macrophages

**DOI:** 10.3390/cells12071009

**Published:** 2023-03-25

**Authors:** You Na Seo, Ji Sue Baik, Song Mi Lee, Ji Eun Lee, Hye Rim Ahn, Min Seo Lim, Moon-Taek Park, Sung Dae Kim

**Affiliations:** 1Research Center, Dongnam Institute of Radiological & Medical Sciences, Busan 46033, Republic of Korea; 2Department of Microbiology and Immunology, College of Medicine, Inge University, Busan 47392, Republic of Korea; 3Department of Medicinal Biotechnology, College of Health Sciences, Dong-A University, Busan 49315, Republic of Korea; 4Department of Veterinary Medicine, College of Veterinary Medicine, Kyoung Pook National University, Daegu 41566, Republic of Korea

**Keywords:** ionizing radiation, DNA damage, p38 MAPK, STAT1, CXCL10, macrophage

## Abstract

Ionizing radiation (IR) is an important means of tumor treatment in addition to surgery and drugs. Attempts have been made to improve the efficiency of radiotherapy by identifying the various biological effects of IR on cells. Components of the tumor microenvironment, such as macrophages, fibroblasts, and vascular endothelial cells, influence cancer treatment outcomes through communication with tumor cells. In this study, we found that IR selectively increased the production of CXC motif chemokine ligand 10 (CXCL10), which is emerging as an important biomarker for determining the prognosis of anticancer treatments, without changing the levels of CXCL9 and CXCL11 in murine J774A.1 macrophages. Pretreatment with KU55933, an ataxia telangiectasia mutated (ATM) kinase inhibitor, significantly inhibited IR-induced CXCL10 production. In contrast, pretreatment with N-acetyl-cysteine or glutathione, a reactive oxygen species scavenger, did not inhibit IR-induced CXCL10 production. Further, we attempted to identify the intracellular molecular target associated with the IR-induced increase in CXCL10 secretion by J774A.1 macrophages. IR phosphorylated p38 mitogen-activated protein kinase (MAPK) and signal transducer and activator of transcription 1 (STAT1) in J774A.1 macrophages, and p38 MAPK and STAT1 were involved in CXCL10 via IR using pharmacological inhibitors (SB203580 and fludarabine, respectively) and the siRNA technique.

## 1. Introduction

Ionizing radiation, in addition to surgery and chemotherapy, is an important tumor treatment modality. Attempts have been made to improve the efficiency of radiotherapy by identifying the various biological effects of IR on cells. Previous studies focused on maximizing the tumor-cell-killing ability of IR by determining its effects on tumor cells [1,2,3]. As various factors in the tumor microenvironment affect tumor cell survival, many studies on the effects of IR on these factors have been conducted recently [4,5,6]. Despite various efforts and scientific advances, information remains limited on predictive and monitoring factors from the tumor microenvironment that influence the efficacy of anticancer radiation therapy.

Components of the tumor microenvironment, such as macrophages, fibroblasts, and vascular endothelial cells, influence cancer treatment outcomes through communication with tumor cells. Among them, macrophages are innate immune cells that account for approximately 50% of tumor mass [7], and studies have shown that various factors secreted by these macrophages affect the prognosis of patients with cancer [8,9,10,11]. In addition, parts that are difficult to explain with the existing M1/M2 paradigm, called tumor-associated macrophages, have been recently reported [12,13,14]. Ultimately, determining the effect on the tumor prognosis is necessary for each radiotherapy case.

CXCL10, also known as interferon γ-induced protein 10 kDa (IP-10) [15], is mainly expressed in immune cells and acts on the CXCR3 receptor of target cells to recruit T lymphocytes and other immune cells [16]. In addition to its role as a chemotactic factor, CXCL10 is involved in cancer cell growth, apoptosis, and angiogenesis [17]. In fact, a close relationship has been reported between the level of CXCL10 and the prognosis of patients with colon adenocarcinoma [18], clear cell renal cell carcinoma [19], and pancreatic adenocarcinoma [20]. However, no report addresses IR’s selective CXCL10 production or the mechanism of production. Furthermore, the source of CXCL10 present in solid tumor tissue has not yet been identified. Therefore, determining the cause and mechanism of CXCL10 production is extremely important for improving the treatment results of patients with tumors.

In this study, we found that IR specifically increased the production of CXCL10 in J774A.1 macrophages. In addition, we attempted to identify the intracellular molecular target involved in the increase of CXCL10 secretion in murine J774A.1 macrophages by IR.

## 2. Material and Method

### 2.1. Cell Culture

Mouse macrophage cell lines J774A.1a and MH-S were purchased from the American Type Culture Collection (Manassas, VA, USA). At 37 °C and in a 5% CO_2_ incubator, J774A.1 and MH-S cells were cultured in RPMI 1640 medium (WelGENE, Daegu, Korea) which was supplemented with 10% heat-inactivated (*v/v*) fetal bovine serum (Gibco, ThermoFisher Scientific, Waltham, MA, USA), penicillin (100 U/mL), streptomycin (100 mg/mL) (Gibco, ThermoFisher Scientific, Waltham, MA, USA), and 0.05 mM β-mercaptoethanol (Sigma-Aldrich, St. Louis, MO, USA).

### 2.2. Cell Irradiation

J774A.1 and MH-S cells were seeded at 2 × 10^5^ cells/mL in a 35 mm plate, irradiated (2 Gy) using a Biobeam8000 (137Cs source) cell irradiator (Gamma-Service Medical GmbH, Leipzig, Germany) at a dose rate of 2.5 Gy/min, and incubated at 37 °C for 48 h.

### 2.3. Immunoblotting

The cells were lysed on ice for 15 min using a RIPA buffer (20-188, Sigma-Aldrich, St. Louis, MO, USA) containing a phosphatase inhibitor cocktail (1861277, Thermo Scientific, Waltham, MA, USA). During lysis, vigorous vortexing was provided once every 5 min, followed by centrifugation (13,000 rpm for 15 min at 4 °C). After transferring the separated supernatant to a fresh tube, a Bradford analysis was used to assess the protein content. For each sample, 30 g of protein was electrophoretically separated on a sodium dodecyl sulfate–polyacrylamide gel. After size-separated proteins were transferred to a nitrocellulose membrane, they were blocked with 5% skim milk (244620, BD Biosciences, Franklin Lakes, NJ, USA) in Tris-buffered saline in Tween (TBST) for 1 h. In the subsequent immunoblotting process, the primary antibody reacted overnight at 4 °C, and the secondary antibody reacted for 1 h at room temperature. Other information is presented in Appendix A. Antibody-coupled proteins were developed through SuperSignal West Pico PLUS Chemiluminescent substrates (34580, ThermoFisher Scientific, Waltham, MA, USA).

### 2.4. Reverse-Transcription Polymerase Chain Reaction (RT-PCR)

RNA was isolated from cells using the RNeasy Plus mini kit (Qiagen, Valencia, CA, USA). Using the iScript cDNA synthesis kit, the cDNA was generated (Bio-Rad, Hercules, CA, USA). The PCR conditions were as follows: 5 min of denaturation at 94 °C, 35 cycles at 94 °C for 30 s, 72 °C at an acceptable loosening temperature for 30 s, and 5 min at 72 °C. On a 2% agarose gel, PCR results were analyzed by electrophoresis. The primers utilized in this work are presented in Appendix A.

### 2.5. Cytokine Array

Cells were cultured in a 35 mm dish, and supernatant liquids were extracted from the irradiated cells. In this experiment, a mouse cytokine array kit panel A (Proteome Profiler Mouse Cytokine Array Kit, R&D Systems, Minneapolis, MN, USA) was used according to the manufacturer’s instructions. Cytokine strength was determined by increasing or decreasing expression in comparison to the control group.

### 2.6. Quantification of the Cytokines (CXCL9, CXCL10, and CXCL11) Secreted from the Macrophage Cells

The culture supernatant from the J774A.1 and MH-S cells, treated as indicated, were collected, and cytokine (CXCL9, CXCL10, and CXCL11) levels were measured using ELISA kits, according to each manual. The CXCL9, CXCL10, and CXCL11 ELISA kits were all manufactured by R&D Systems (Minneapolis, MN, USA), and their respective catalog numbers are DY492, DY466, and DY572.

### 2.7. Flow Cytometry Analysis of ROS Production

J774A.1 cells (2.5 × 10^5^ cells/mL) were incubated 48 h after irradiation. The remaining culture medium was removed, and the cells were washed with PBS. Intracellular ROS levels were measured using the fluorescent dye 2′,7′-dichlorodihydrofluorescein diacetate (DCFDA; Sigma-Aldrich, St. Louis, MO, USA), followed by treatment with 1.25 mM of DCFDA in a dark room for 30 min.

### 2.8. siRNA Transfection

J774A.1 cells were transfected with *scramble* siRNA and mouse *STAT1* siRNA (Thermo Fisher Scientific, Waltham, MA, USA), using Lipofectamine RNAiMax transfection reagent (Invitrogen, ThermoFisher Scientific, Waltham, MA, USA) in compliance with the manufacturer’s directions.

### 2.9. Statistical Analyses

Three or more separate experiments were examined to statistically compare the experimental groups. SPSS, version 18.0, was used to investigate the statistical significance between the groups using Student’s *t*-test (SPSS, Chicago, IL, USA). More than two groups were analyzed with a one-way ANOVA (GraphPad Prism version 8.0, GraphPad Software Inc., San Diego, CA, USA). All statistical values were expressed precisely as they were, except when the *p*-value < 0.0001. When the *p*-value < 0.05, statistical significance was determined.

## 3. Results

### 3.1. IR Selectively Increases Only CXCL10 among the CXCL9/CXCL10/CXCL11 Groups

To investigate whether IR affects cytokine release in macrophages, we performed a cytokine array in murine J774A.1 macrophages. After 2 × 10^5^ cells/mL of J774A.1 cells were irradiated, the supernatant was collected 2 days later and subjected to a cytokine array. As shown in Appendix A, the secretion of CXCL10 was increased most in the irradiated macrophages (10.68 ± 0.008 fold). Furthermore, the production of cytokines such as Chemokine (C-C motif) ligand 5 (CCL5) and interleukin 1 receptor antagonist (IL-1ra) in irradiated J774A.1 macrophages increased more than twice compared to the control J774A.1 macrophages. The total cytokine array results are tabulated in Appendix A.

To validate the cytokine array’s results, a sandwich ELISA was performed on the supernatant of the irradiated J774A.1 cells. Two days after irradiating 2 × 10^5^ cells/mL of J774A.1 cells with the indicated dose, the supernatant was collected and a sandwich ELISA was performed. As shown in Figure 1A, the IR dose-dependently increased CXCL10 secretion in murine J774A.1 macrophages. Thereafter, we used a dose of 2 Gy for the irradiation experiment using murine J774A.1 macrophages. The same procedure was followed with murine MH-S macrophages. As shown in Figure 1B, irradiating murine MH-S macrophages according to the indicated dose also dose-dependently increased CXCL10 secretion. Experimental results using two macrophages confirmed that the irradiation increased CXCL10 secretion in both J774A.1 cells and MH-S cells. In addition, we confirmed that the IR increased the mRNA level of *CXCL10* in both macrophage cell lines (Appendix A).

We next confirmed the pattern of CXCL10 secretion over time after radiation exposure. As shown in Figure 1C,D, CXCL10 production did not increase until 1 day after irradiation. After 2 days of irradiation, a significant increase in CXCL10 production was observed in both J774A.1 and MH-S cells.

CXCL9, CXCL10, and CXCL11 are CXC chemokines that share the CXCR3 receptor. Reports have asserted that the CXCL9, CXCL10, and CXCL11 axes are important for immune activation in cancer therapy [21]. As CXCL9, CXCL10, and CXCL11 are often generated together by various stimuli, a sandwich ELISA was employed to confirm whether IR increased the production of these three chemokines. As shown in Figure 1E, CXCL9 and CXCL11 did not increase considerably; however, as shown in the previous Figure 1B, IR increased CXCL10 production significantly in J774A.1 macrophages. Additionally, among CXCL9, CXCL10, and CXCL11, only CXCL10 was secreted at a measurable level in the control J774A.1 macrophages.

### 3.2. Effect of DNA Damage by IR on CXCL10 Production in J774A.1 Macrophages

Another significant biological effect of IR on cells is DNA damage. Therefore, we investigated whether IR alters the expression of DNA-damage-related genes in the J774A.1 macrophage cell line under our experimental conditions and whether the pharmacological inhibitor KU55933 (10 µM) blocks this change in gene expression. The ATM inhibitor KU55933 was pretreated 30 min before J774A.1 cells were exposed to IR. As shown in Figure 2A, IR increased the phosphorylation of genes related to DNA damage, such as H2AX, CHK1, and ATM, in J774A.1 macrophages. In addition, pretreatment with KU55933 inhibited the increased phosphorylation of DNA-damage-related genes caused by IR in J774A.1 macrophages. In parallel with measuring the expression and phosphorylation of intracellular DNA-damage-related proteins, an experiment was conducted to measure secreted CXCL10 using the supernatant of J774A.1 macrophages. Figure 2B shows that KU55933 inhibits an IR-induced increase in CXCL10 secretion.

### 3.3. STAT1 Is Involved in Increased CXCL10 Secretion in J774A.1 Macrophages by IR

STAT1 is involved in increased CXCL10 production [22]. As shown in Figure 3A,C, IR increased both the expression and phosphorylation of the STAT1 protein in J774A.1 macrophages. In addition, both the protein expression and phosphorylation of STAT1 by IR were decreased by the STAT1 inhibitor fludarabine (50 µM). The STAT1 inhibitor fludarabine was pretreated for 30 min before the J774A.1 cells were exposed to IR. The simultaneously conducted sandwich ELISA of CXCL10 confirmed that fludarabine, a STAT1 inhibitor, significantly inhibited increased CXCL10 production by IR (Figure 3B).

Next, we investigated whether IR changed the mRNA expression of STAT1 in J774A.1 macrophages. As shown in Appendix A, the results confirmed that IR increased both the mRNA expression and protein production of STAT1.

### 3.4. Among the Three Types of MAPKs, p38 MAPK Is Involved in the IR-Induced Increase in CXCL10 Secretion in J774A.1 Macrophages

Next, we investigated whether the MAPK signaling pathway of J774A.1 macrophages is changed by IR and is involved in increased CXCL10 secretion. Figure 4A shows that IR increased the phosphorylation of p38 MAPK and ERK in J774A.1 macrophages. Using pharmacological inhibitors for each MAPK (PD98059 for ERK, SB203580 for p38 MAPK, and SP600125 for JNK), we investigated which subtype of MAPK phosphorylation was involved in the increase in CXCL10 secretion by IR. Among the three MAPK subtype inhibitors, only 2.5 µM of SB203580 inhibited IR-induced CXCL10 production (Figure 4B). Figure 4C shows that SB203580 also inhibited the IR-induced phosphorylation of p38 MAPK. These results suggest that p38 MAPK is involved in the IR-induced increase in CXCL10 secretion in J774A.1 macrophages. In addition, SB203580 inhibited the increase in STAT1 protein expression and phosphorylation by IR. These results suggest that phosphorylation precedes the increase in phosphorylation and expression of STAT1 protein.

## 4. Discussion

Ionizing radiation (IR) is an effective treatment method that inhibits the growth of tumor cells by directly inducing DNA damage to tumor cells. In addition, IR shrinks cancer cells and causes various pathophysiological changes in components constituting the tumor microenvironment [23,24,25]. In particular, the effect of IR on macrophages, which occupy a large portion of tumor tissue, has been reported [26,27,28]. IR causes macrophages to produce more IL-1β in response to LPS [29,30]. As such, various, sometimes contradictory, reactions to IR occur under different pathophysiological conditions. To the best of our knowledge, this study clearly discovered, for the first time, that IR selectively produces CXCL10 in macrophages.

CXCL10 production is usually expressed in the same way as chemokines that share a receptor called CXCR3, such as CXCL9 and CXCL10 [31]. In our experimental results, however, IR increased only the production of CXCL10 (Figure 1E). There are reports on the correlation between the prognosis of tumor patients and the level of CXCL10. Essentially, these reports were based on the intrinsic ability of CXCL10 to recruit immune cells such as cytotoxic T cells and natural killer cells. In metastatic melanoma, intratumoral CXCL10 exhibited a good prognosis by recruiting T cells [32]. In another example, the HeLa cervical cancer cell xenograft model showed a synergistic effect when CXCL10 gene therapy and radiotherapy were combined [33]. Additionally, 2 Gy of IR increased T lymphocyte recruitment by increasing CXCL9 and CXCL10 levels in AGS-EBV human gastric cancer cells [34]. In clinical studies, CXCL9 and CXCL10 expression levels showed a high correlation with the results of immunotherapy using PD-1 blockade. In contrast, some reports have indicated a high correlation between CXCL10 production and poor tumor prognosis. CXCL10, produced by IFN-γ, is involved in radioresistance through CXCR3 in Hep2 cells [35]. CXCL10 also has been clinically reported to be useful as a significant biomarker in assessing cardiac function in children infected with SARS-CoV-2 [36].

Various reports have addressed the source of CXCL10 in the tumor microenvironment. For example, CXCL10 is produced by immune cells and melanoma cells in human melanoma metastases [32]. Irradiation increased CXCL10 production in MC38 mouse colon adenocarcinoma cells [37]. Additionally, in an in vivo experiment using the same MC38 tumor xenograft model, the production of IFN-γ and CXCL10 in irradiated tumor tissue increased [38]. These reports suggest that CXCL10 production sources vary according to stimuli, and that CXCL10 may be secreted from both tumor and immune cells. As the effect on tumor treatment may vary depending on the type or source of CXCL10 generation stimulation, specifying the stimulation and generation source is very important. Our findings suggest that if CXCL10 production is specifically increased during radiotherapy, its source may be a macrophage. To the best of our knowledge, this is the first report on a cellular source that specifically generated CXCL10 in response to radiotherapy.

IR-induced pathophysiological changes in macrophages have been reported in several studies. When confirmed through culture media of primary human monocyte-derived macrophages (MDM)s irradiated with 5 Gy, the secretion of proinflammatory cytokines such as CXCL10, CCL2, and IL-6 increased. The supernatant of 5 Gy-irradiated THP-1 cells also increased the secretion of various proinflammatory cytokines such as IL-1β, IL-6, TNF-a, IL-10, and CCL2 [39]. IR increased the production of proinflammatory cytokines, especially CXCL10, IL-6, and CCL2, in human MDMs and THP-1 cells. In our study, only the production of CXCL10 was selectively increased, and IR did not increase IL-6 production in murine J774A.1 macrophages. This is most likely due to a difference in the characteristics of each cell line. Therefore, a future study will be needed to identify and compare the characteristics affecting these differences.

The transcription factor STAT1 has been associated with CXCL10 in various pathophysiological situations. For example, STAT1 is involved in the induction of CXCL10 in hepatocyte lipotoxicity [40]. STAT1 is also involved in the increased production of CXCL10 by IL-1β and IFN-γ in rat islets and β cell lines [41]. Furthermore, STAT1 and CXCL10 affect bone destruction through the M1 polarization of macrophages [22]. CXCL10 and STAT1 have also been suggested as key candidate genes for understanding the molecular mechanism of melanoma [42]. STAT1 is known to be activated by IFN-γ. However, in our study, J774A.1 macrophage cells did not produce IFN-γ at a level at which IR could be measured (Appendix A). This result implies that IR-induced CXCL10 production in J774A is independent of IFN-γ. In addition, IR not only increased the phosphorylation of the STAT1 protein in J774A.1 cells but also significantly increased protein expression (Figure 3A). Moreover, IR increased the expression of STAT1 mRNA in J774A.1 macrophages (Appendix A). In terms of CXCL10 production, this means that IR can replace the action of IFN-γ in J774A.1 macrophages.

There have been few reports on the increase in STAT1 protein expression caused by IR in macrophages. In a Raw264.7 macrophage cell line, ultraviolet B (UVB) radiation increased phosphorylation of the STAT1 protein and suppressed NOX2 expression by IFN-γ [43]. It also has been reported that UV irradiation increased the phosphorylation of STAT1 protein at 40 J/m^2^ in Bac1.2F5 macrophages [44]. However, the basic expression level of STAT1 protein in the control group was high, and the fact that only phosphorylation increased, without an increase in STAT1 protein expression by UV irradiation, was different from our study results. This is probably due to the different cell lines used (Bac1.2F5 macrophage cell vs. J774A.1 macrophage cell) and the different stresses used (UV irradiation representing daily radiation exposure vs. ionizing radiation representing medical radiation exposure). To the best of our knowledge, this is the first study to demonstrate that IR increases both STAT1 mRNA and protein levels in macrophages. In addition to the macrophage cell line, IR increased both the expression and phosphorylation of the STAT1 protein in A549 lung adenocarcinoma cells [45].

p38 MAPK, an essential regulator of proinflammatory cytokine release in macrophages, is also involved in the activation of the STAT1 protein [46]. According to one study, p38 MAPK is involved in the phosphorylation of STAT1 by interferons in HeLa cells [47]. TNF-α promotes epithelial-–mesenchymal transition by stimulating the CXCL10/CXCR3 axis of colorectal cancer cells. In this process, the p38 MAPK, NF-κB, and phosphoinositide 3-kinase (PI3K)-Akt signaling pathways are involved in the secretion of CXCL10 by TNF-α [48]. The production of CXCL10 by prolactine and interferon (IFN)-γ in keratinocytes is inhibited by p38 MAPK inhibitor SB203580 and the extracellular signal-regulated kinase (ERK) inhibitor PD98059 [31].

It has been reported that IR increases the phosphorylation of both ERK and p38 MAPK proteins but not Jun N-terminal kinase (JNK) in A549 lung adenocarcinoma cells [45]. In our experimental results, IR also increased the phosphorylation of ERK in J774A.1 macrophages (Figure 4A). However, PD98059, an ERK-specific inhibitor, failed to suppress IR-induced CXCL10 production (Figure 4B). Therefore, depending on the cell type (J774A.1 cell) and the stimulus type (IR) the cell is provided, the selective production of CXCL10 and the production mechanism may vary. Another possibility is that in the pathogenesis of rheumatoid arthritis, CXCL10 acts on CXCR3 rather than TLR4 on macrophages or T lymphocytes to activate ERK, thereby increasing the ability of macrophages or T lymphocytes to migrate [49]. This possibility is also supported by a previous report that CXCL10 knockdown inhibits the activation of ERK signaling in murine mesangial cells [50].

However, this study had a limitation in that increased CXCL10 production by IR was confirmed only in the J774A.1 cell. To overcome this limitation, we found that IR increased CXCL10 production in MH-S macrophages (Figure 1B,D), and our results were not limited to the J774A.1 macrophage cell. In our subsequent study, we will use mouse-derived macrophages to investigate the efficacy of irradiated macrophages in tumor treatment using an in vivo model. Further, it should be investigated whether an IR-induced increase in CXCL10 production is observed in human-derived macrophages. As J774A.1 macrophages exhibit properties of both monocytes and M0-type macrophages, studying an IR-induced increase in CXCL10 production in human THP-1 macrophages, which have a well-characterized subtype, will be more clinically relevant.

## 5. Conclusions

For the first time, we report that CXCL10 is the only cytokine in the CXCL9/CXCL10/CXCL11 groups that can be detected and induced by IR via the DNA-damage-induced p38 MAPK-STAT1 signaling pathway in the macrophage cell lines.

## Figures and Tables

**Figure 1 cells-12-01009-f001:**
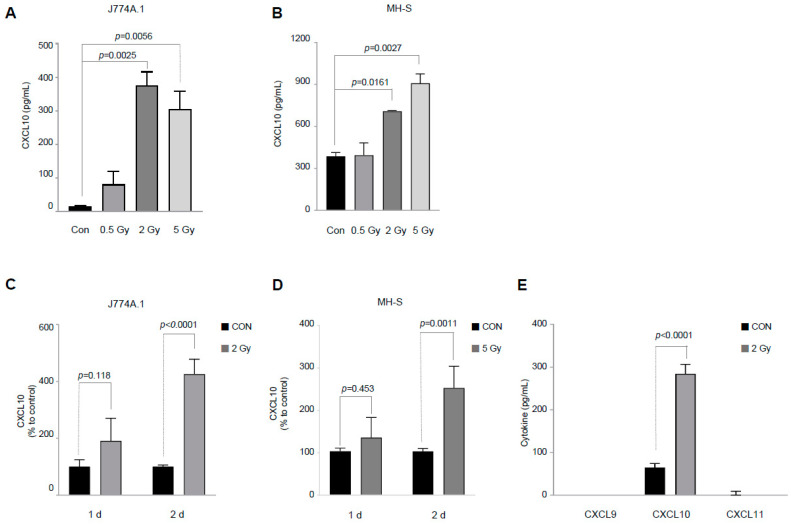
Ionizing radiation induces CXC ligand 10 (CXCL10) production in both the J774A.1 macrophage and MH-S macrophage cell lines. (**A**,**B**) Forty-eight hours after exposure to ionizing radiation at the indicated dose for the the 2 × 10^5^ cells/mL of J774A.1 (**A**) and MH-S (**B**) murine macrophage cell lines, the amount of CXCL10 released into the cell culture medium was measured (n = 3–4). (**C**,**D**) After the J774A.1 (**C**) and MH-S (**D**) macrophages were irradiated with 2 Gy and 5 Gy, respectively, changes in CXCL10 production were observed over time. To compare the relative increase in CXCL10 production, it was expressed as relative production levels compared to each control group (n = 5–7). (**E**) The amounts of CXCL9, CXCL10, and CXCL11 released into the cell culture medium were measured 48 h after the J774A.1 macrophage cell line at 2 × 10^5^ cells/mL was exposed to 2 Gy of radiation (n = 3). Bar graphs are presented as the mean ± standard error of the mean value.

**Figure 2 cells-12-01009-f002:**
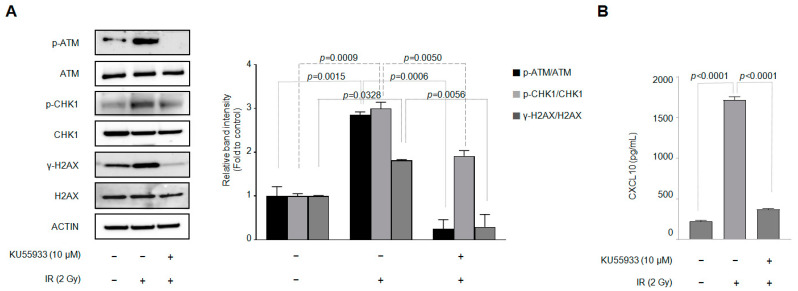
DNA damage by ionizing radiation is involved in the production of CXCL10 in J774A.1 macrophages. A total of 2 × 10^5^ cells/mL of the J774A.1 macrophage cell line, with or without the ATM kinase inhibitor KU55933 (10 μM), was exposed to 2 Gy of radiation. Forty-eight hours later, (**A**) the DNA damage-related protein expression and (**B**) the amount of CXCL10 released into the cell culture medium were measured. KU55933 (10 μM) was pretreated 30 min before irradiation (n = 3). Bar graphs are presented as the mean ± standard error of the mean value.

**Figure 3 cells-12-01009-f003:**
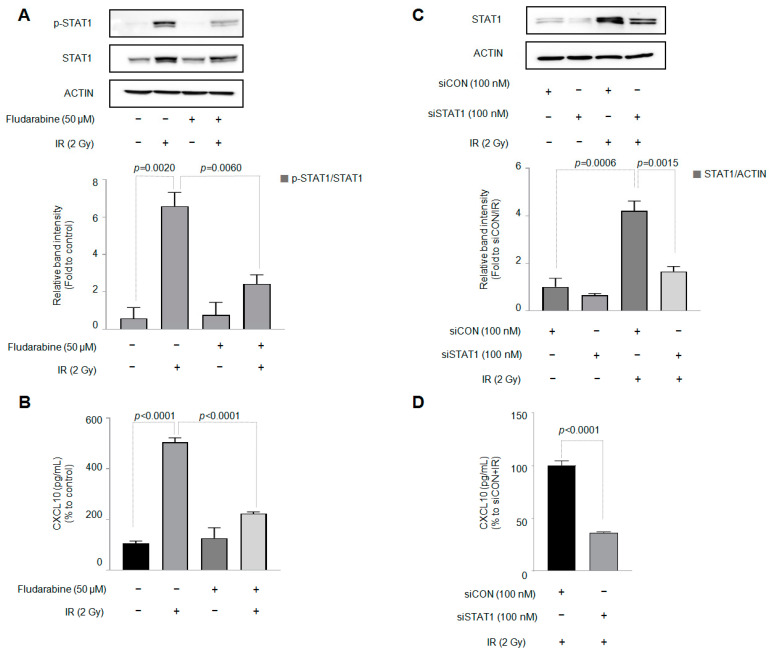
Activation of the signal transducer and activator of transcription 1 (STAT1) protein by ionizing radiation is involved in the production of CXCL10 in J774A.1 macrophages. A total of 2 × 10^5^ cells/mL of the J774A.1 macrophage cell line, with or without the STAT1 inhibitor fludarabine (50 μM), was exposed to 2 Gy of radiation. Forty-eight hours later, (**A**) the total STAT1 and phosphorylated STAT1 protein expression and (**B**) the amount of CXCL10 released into the cell culture medium were measured. Fludarabine (50 μM) was treated 30 minutes before the cells were exposed to radiation (n = 3). (**C**) After knocking down the STAT1 gene in J774A.1 macrophages, using the siRNA technique, the production of CXCL10 was confirmed (n = 3). (**D**) The knockdown efficiency of STAT1 was validated (n = 3). Bar graphs are presented as the mean ± standard error of the mean value.

**Figure 4 cells-12-01009-f004:**
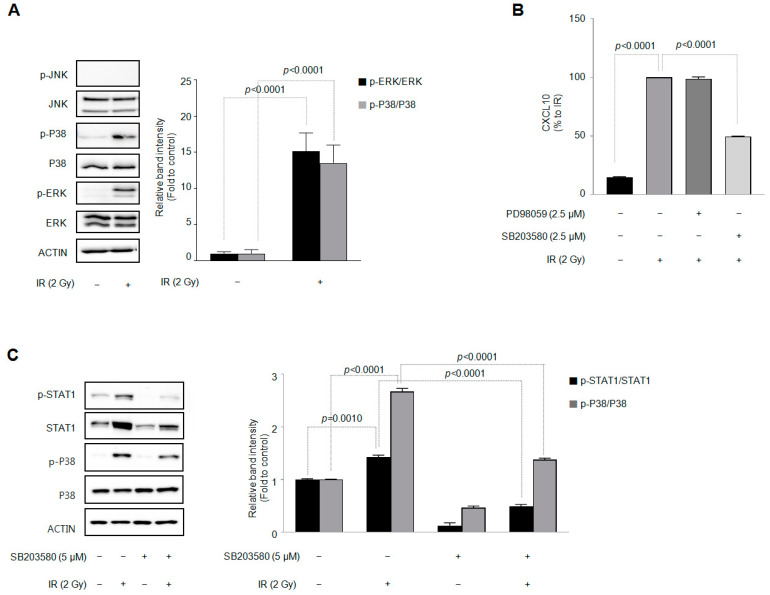
p38 MAPK activation by ionizing radiation is involved in the production of CXCL10 in J774A.1 macrophages. A total of 2 × 10^5^ cells/mL of the J774A.1 macrophage cell line, with or without three MAPK subtype-specific inhibitors, namely, PD98059 (2.5 μM) for ERK, SP600125 (2.5 μM) for JNK, and SB203580 (2.5 μM) for p38 MAPK, was exposed to 2 Gy of radiation. Forty-eight hours later, (**A**) the expression of the three MAPK subtypes and (**B**) the amount of CXCL10 released into the cell culture medium were measured. Each MAPK subtype inhibitor was pretreated 30 min before irradiation (n = 3–4). (**C**) Crosstalk between activated p38MAPK and activated STAT1 in irradiated J774A.1 macrophages (n = 3). Bar graphs are presented as the mean ± standard error of the mean value.

## Data Availability

Not applicable.

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
