# Peer review of "Ionizing Radiation Selectively Increases CXC Ligand 10 Level via the DNA-Damage-Induced p38 MAPK-STAT1 Pathway in Murine J774A.1 Macrophages"

_cells, 2023, doi:10.3390/cells12071009_

Round 1

Reviewer 1 Report

In this manuscript, the authors found that IR induced the p38/stat1-mediated expression of CXCL10 in mouse macrophage cell lines. The data are clear to draw conclusions and the material and methods were clearly shown. The manuscript was well-written and easy to follow.

One minor point is that the conclusion of Fig2 needs to be modified. Based on the data provided by the authors, CXCL9 and CXCL11 are barely detected in the macrophage cell line and these two proteins were not induced by the IR. So it is not accurate to claim that IR selectively increases only CXCL10 among the CXCL9/CXCL10/CXCL11 groups. The conclusion should be CXCL10 is the only cytokine in the CXCL9/CXCL10/CXCL11 groups that could be detected and induced by IR in the mouse macrophage cell lines.

Author Response

Dear Reviewer 1

We would like to appreciate the constructive advice of the reviewers greatly. 

The revisions to the reviewer's suggestions are presented point-by-point in the response letter.

Sincerely yours

Sung Dae Kim

Reviewer 2 Report

Manuscript title: Ionizing radiation selectively increases CXC ligand 10 level via the DNA damage-induced p38 MAPK-STAT1 pathway in murine J774A.1 macrophages.

In this manuscript, authors found that ionizing radiation (IR) selectively increased the production of CXC motif chemokine ligand 10 (CXCL10) through a cytokine array in murine J774A.1 macrophages. They also noticed DNA damage upon IR represented by increase of phospho-ATM. Furthermore, they observed that p38 MAPK and STAT1 were involved in increase of CXCL10 by IR in J774A.1 macrophages by using inhibitors and siRNA. This is a good start for translation. However, I have a few comments that need further attention prior to resubmission.

1.     Some expression needs to be modified, like line 52 on page2. As a chemokine, CXCL10 can not be secreted by another cytokine.

Line 85-86, the protein was transferred to the buffer. It does not make sense.

There are also some other places, the expression is confusing, please modify accordingly.

2.     Method 2.7 line 116-127, you described the experiment cell cycle analysis. But I did not see this data in your text or figures. Please delete this method or add the related results.

3.     For the statistic analysis, if your data is more than 2 groups, student’s t-test is now suitable anymore. You should use one-way ANOVA analysis here. Please verify with your statistic expert and update it in your paper accordingly.

4.     In figure 1B, the p value of comparing the concentration of CXCL10 in column 5Gy with control group is 0.0093, which is significant. But in figure 1D, the p is 0.086 at 2d (not significant). How to explain this?

5.     I think you could merge your figure 2 with figure 1, which also means that you merge results 3.1 and 3.2. As figure 2 is a supplemental verification of your cytokine array. We can see that IR has no effect on CXCL9 and CXCL11 from supplementary figure 1B.

6.     I highly recommended moving results 3.3 and 3.6 to the supplementary file. They are the negative results supporting your hypothesis. If you put them in the main figures, they kind of affected the fluency and integrity of the paper.

7.     Pay attention to the sequence of supplementary figure. Supplementary figure 4 to result 3.4, supplementary figure 3 to result 3.5. A little confusing.

8.     In supplementary figure 4, the quantification of p-H2AX/H2AX in KU55933(+)/IR (+) group is not significant compared to control. How to explain it? In figure 4A, from your IB bands, I noticed that total H2AX was reduced in KU55933(+)/IR (+) group, but p-H2AX/H2AX is not reduced in KU55933(+)/IR (+) group. These data is not consistent with your text.

9.     The column figure in Figure 5A has the same meaning as figure 5B. To me, they are just 2 different analysis methods of 1 experiment. Please keep only one in the main figure.

This is the same case for Figure 5C and 5D.

10.  In figure 7A, the p value for p-P38/P38 is 0.0713, whish is considered not significant. It is not consistent with your results described in your text. Please address it. And you also missed the error bar in p-P38/P38 IR (+) column.

And in figure 7B, if there is no expression of p-JNK in samples with or without IR, it is meaningless to put the JNK inhibitor data in it.

11.  To clarify, in Figure 6, LY294002 is a PI3k inhibitor, not an inhibitor specific to AKT. Please also discuss why LY294002 increased CXCL10 in LY294002(+) IR (+) group.  

From the description about SB203580 (https://www.selleckchem.com/products/SB-203580.html), SB203580 also blocks PKB phosphorylation with IC50 of 3-5 μM. In your paper, you used 5μM SB203580. Is it possible that the change of p-STAT1 is mediated by inhibition of PKB (AKT), not just P38. Did you check p-AKT and AKT here in figure 7?

Have you tried AKT specific inhibitor, like LY2780301?

     12.     What is the status of DNA damage in MH-S cells? And what is the                          expression of P-STAT1/STAT1 (RNA level in supplementary figure 3 is not             enough) and p-P38/P38 in MH-S? How about the change after using            inhibitors?

13.     Line 406, MH-S macrophages (Figures 1C and 1D) should be MH-S macrophages (Figures 1B and 1D). 

1

Author Response

Dear Reviewer 2

We would like to appreciate the constructive advice of the reviewers greatly. 

The revisions to the reviewer's suggestions are presented point-by-point in the response letter.

Sincerely yours

Sung Dae Kim

Round 2

Reviewer 1 Report

The current manuscript is good to publish

Author Response

Response to Reviewer 1

We want to thank the reviewers for their effort again.

The authors carried out the reviewer's two suggestions during this second round review process. The reviewer's effort could further develop the author's manuscript.

We would like to appreciate it again.

Sincerely yours,

Sung Dae Kim

Reviewer 2 Report

The authors have addressed most of my concerns and significantly improved the manuscript. Thanks.

Here, I have 2 minor suggestions.

1.     Figure 3 in cover letter is different from figure 3 in revised manuscript. Please verify it. And please check it is STAT1/P-STAT1 or P-STAT1/STAT1, too.

2.     As to my previous comment 10, I mean it is better to delete the data related to JNK in figure 7B which showed the CXCL10 amount after using JNK inhibitor SP600125 (2.5 μM). As in 7A, there is no expression of p-JNK upon IR.

Author Response

We want to thank the reviewers for their effort again.

The authors carried out the reviewer's two suggestions during this second round review process. The reviewer's effort could further develop the author's manuscript.

We would like to appreciate it again.